# Qualitative Analysis of a Twitter-Disseminated Survey Reveals New Patient Perspectives on the Impact of Urinary Tract Infection

**DOI:** 10.3390/antibiotics11121687

**Published:** 2022-11-23

**Authors:** Marissa Valentine-King, Lindsey Laytner, Casey Hines-Munson, Kiara Olmeda, Barbara Trautner, Sheryl Justice, Christina Ching, Larissa Grigoryan

**Affiliations:** 1Department of Family and Community Medicine, Baylor College of Medicine, Houston, TX 77098, USA; 2Center for Innovations in Quality, Effectiveness and Safety (IQuESt), Michael E. DeBakey Veterans Affairs Medical Center, Houston, TX 77021, USA; 3Department of Medicine, Section of Health Services Research, Baylor College of Medicine, Houston, TX 77030, USA; 4College of Nursing, The Ohio State University, Columbus, OH 43210, USA; 5Nephrology and Urology Research Affinity Group, Nationwide Children’s Hospital, Columbus, OH 43205, USA; 6The Kidney and Urinary Tract Center, The Abigail Wexner Research Institute at Nationwide Children’s Hospital, Columbus, OH 43215, USA; 7Department of Pediatric Urology, Nationwide Children’s Hospital, Columbus, OH 43205, USA

**Keywords:** urinary tract infections, qualitative research, social media, quality of life, mental health, patient care, antibiotics, biofilms

## Abstract

Few studies have harnessed social media to explore patients’ experiences with urinary tract infection (UTI); therefore, we captured UTI experiences and future research suggestions through a Twitter-disseminated survey. The survey posed three qualitative questions inquiring about the impact of UTIs, greatest UTI management hurdle, and research suggestions. We also asked participants to rate how seriously others perceive UTIs and the importance of UTIs in their life (scale: 1–100 (highest)). The study period spanned from January to June 2021. Coding was performed in duplicate, followed by thematic analysis. Of 466 participants from 22 countries, 128 considered their UTIs recurrent (*n* = 43) or chronic (*n* = 85). Six major themes emerged: UTIs drastically impact (1) physical and (2) mental health and (3) cause severe limitations in life activities. Patients reported (4) negative clinician interactions and perceived inadequate care, (5) a lack of knowledge and awareness surrounding UTIs, and (6) research gaps in UTI diagnostics and treatment. The participants considered UTIs extremely important (median: 100, IQR: 90–100), but characterized others’ perceptions of them as less serious (median: 20, IQR: 10–30). Our survey revealed a patient population struggling with UTIs, particularly chronic UTIs. Our findings highlight perceived shortcomings in current UTI treatment and diagnostics.

## 1. Introduction

Urinary tract infections (UTIs) cause substantial morbidity worldwide, with the global burden estimated at 404.6 million cases in 2019 [1]. The impact of UTIs on females is considerable, as 60% of women have reported experiencing a UTI throughout their lifetime [2]. UTIs also exert significant impacts on quality of life, accounting for 5.2 million disability-adjusted life years (DALYs) in 2019, globally [1]. Antibiotic resistance among uropathogens is also increasing; for example, Gram-negative urinary organisms collected from outpatients across all regions of the United States (US) now have antimicrobial resistance levels above the thresholds recommended for empiric UTI treatment [3]. With rising antimicrobial resistance contributing to increased disability and deaths among bacterial infections globally, the burden of UTIs on society will likely increase [1,4]. Indeed, Zeng et al. detected an increase in age-adjusted UTI incidence and mortality between 1990 and 2019 by 514 cases and 1.3 deaths per 100,000 individuals, respectively [1,4]. To mitigate rising resistance, the field of antibiotic stewardship has been established. Antibiotic stewardship interventions in the context of UTIs aim to ensure the following: providers make the right diagnosis; prescribe the correct, empiric antimicrobial at an appropriate dose and duration; and de-escalate based on urine culture susceptibilities and/or when negative [5].

A considerable number of patients develop recurrent UTI, defined as two culture-proven episodes of bacterial cystitis with symptoms within 6 months or three within 12 months [6,7,8,9,10]. Prospective studies have identified 24% of college-aged females, 44% of middle-aged women, and 13% of older males in outpatient settings developed recurrent UTIs after an initial index episode [6,7,8,9]. UTI recurrence in premenopausal women has been associated with substantial impacts on quality of life from bodily pain and emotional, social, and mental health impacts [11]. In addition, over 80% of women with recurrent UTIs reported pathologic scores on the Female Sexual Function Index and the Female Sexual Distress Scale [12]. Urological societies and/or governments in the US, Canada, Australia, Europe, and the UK have published recurrent UTI treatment and management guidelines, most of which are targeted towards treatment of uncomplicated, recurrent UTI in women [10,13,14]. All include dosing for prophylactic antibiotics, and all but the UK guidelines endorse obtaining a urine culture for initial diagnosis [10,13,14]. 

Few qualitative studies have examined patients’ experiences with recurrent UTIs [15,16,17,18]. Most of these studies have utilized traditional interviews or focus groups, which may limit freedom of responses due to the sensitive nature of the topic [15,16,18]. However, one study that analyzed content on an Internet-based support forum, hosted by the UK–based Cystitis and Overactive Bladder Foundation, revealed themes centered on atypical symptomology, pervasive impacts of UTI on many aspects of life, short-lived symptom suppression from antibiotics, and mixed experiences with clinicians [17]. The authors credited the level of participant disclosure to the disinhibiting effects and honesty that in-person interviews may not facilitate [17,19]. However, this study was limited to participants who knew of the UK-hosted forum and may only represent patient experiences in the UK. Furthermore, the forum could only examine patient-driven conversations, and researchers could not ascertain demographic details therein. Thus, the use of an Internet platform with international reach for survey dissemination would allow for a wider catchment of patients, not limited to a particular practice, health network, or country, which could improve the diversity of responses or health experiences, or detect sequelae that go unnoticed in clinical practice. 

We aimed to capture patient experiences using a Twitter-disseminated survey to reach a wider portion of the global population and facilitate disclosure that traditional interviews may not foster. Our survey included three qualitative questions that inquired about the impact of UTIs, UTI management challenges, and suggestions for future research from patients, caregivers, and clinicians. We also included a quantitative poll to gauge how participants rated the importance of UTIs in their lives and their perception of how others viewed the seriousness of their illness. Within this study, about a quarter of patients noted they had either recurrent or chronic UTIs, which imparted considerable physical and mental suffering, and revealed a perceived lack of support from clinicians. Participants expressed feelings that UTIs are often trivialized by healthcare providers and the public. Lastly, participants highlighted shortcomings in UTI research and emphasized the need for improved diagnostics, changes in national guidelines, and safer treatments to alleviate their suffering from chronic UTIs. 

## 2. Results

### 2.1. Population Demographics, Survey Response Rate, and Twitter Analytics

We captured responses from 466 participants from 22 countries during a 6-month time frame. As described in the methods, this initial survey did not capture any demographic information outside of the country of origin. Out of 448 participants who provided their email address for further contact, 207 (44.4%) responded to our follow-up demographic poll, which revealed a population composed primarily of White (95%) females (97%). The age distribution was more equally dispersed, with more respondents between 41 and 60 years (39%), followed by 33% between 61 and 80 years, and 28% between 18 and 40 years (Table 1). The majority of the survey participants were patients with UTIs (*n* = 203); one considered themself a patient and a researcher, and the remaining participants (*n* = 3) were either family members or friends. No participants self-identified as clinicians. The majority of the participants (55.2%, *n* = 257) were from the UK or Ireland, followed by the US or Canada (27.3%, *n* = 127), Oceania (Australia or New Zealand) (11.2%, *n* = 52), or Europe (5.4%, *n* = 25) (Table 1).

Out of 466 submissions, over 98% of the participants provided responses to the three open-ended, qualitative questions: Q1: How is your life affected by UTIs (*n* = 460)? Q2: What do you think is the greatest hurdle in managing UTIs (*n* = 464)? Q3: What would you say to researchers studying UTIs (*n* = 464)? There was a similarly high participation rate (>98%) for the quantitative poll questions, wherein 458 participants provided a rating for “How seriously do you think UTIs are taken by others?” and 464 participants provided a rating for “How important are UTIs in your life?”.

In terms of Twitter analytics, there were 5002 impressions, 37 likes, and 29 retweets. During the same time frame, the posting incurred 302 engagements (the number of times a user interacted with a Tweet). This metric includes retweets and likes, but also the following: replies, follows, links, cards, hashtags, embedded media, or Tweet expansion. Groups that liked/retweeted included the following: “Fibroid New Research”, “Sling the Mesh”, “Enabledlife”, and “Tractivus”. Two groups, the “Chronic Urinary Tract Infection Campaign” and “Chronic UTI Australia”, independently sent out unprompted, separate Tweets to encourage survey participation.

### 2.2. UTI Relationship and Type 

Although our survey questions did not mention or ask specifically about “chronic UTI”, 128 participants reported having a chronic or embedded UTI (*n* = 85) or recurrent UTIs (*n* = 43). The phenotype of chronic or embedded UTI was described as having constant infections that lasted for weeks, months, or years. Others described experiencing recurrent UTIs, which had been previously treated successfully until their subsequent UTI became refractory to treatment and caused constant UTI symptoms. The following excerpt reflected several common components experienced by sufferers of chronic UTI (will further be referred to as chronic UTI), in terms of the diagnosis and symptomology:


*“This year of my life has been hell. I’ve had a chronic, embedded UTI infection from an acute infection in October 2019. The infection was resistant to trimethoprim and kept recurring. Since March 2020 [presently 1/2021] symptoms became constant, yet my urine cultures no longer showed bacteria. I had severe UTI symptoms, stinging, burning, frequency, urgency, and it has changed my life. I no longer could work. My doctors diagnosed me with interstitial cystitis, but I knew it was a UTI as antibiotics were the only thing to bring me relief. It spread to my kidneys over Christmas, and I had a kidney infection, which was painful. I’m now seeing urologist Dr. X and he did a DNA sequencing test on my urine and found the hidden bacteria and diagnosed me with a chronic, embedded UTI”.*

*(New Zealand, 123568474)*


### 2.3. Thematic Analysis

Thematic analysis revealed six prominent themes (Figure 1). The first five themes stemmed primarily from responses from qualitative questions Q1 and Q2, which inquired about how participants’ lives were affected by UTI and their greatest UTI management challenges. The sixth theme was based upon responses to Q3, which asked about research suggestions.

The major themes are as follows: (1) UTIs negatively impact physical health, (2) drastically affect mental health, and (3) cause severe limitations to many aspects of life. (4) Patients also reported major shortcomings in UTI treatment and management, accompanied by negative clinician interactions and barriers in accessing a chronic UTI specialist. (5) When patients did seek care, they perceived that clinicians lacked scientific knowledge and awareness of UTI and chronic UTI and trivialized the disease experience. (6) Lastly, participants revealed that research is desperately needed, especially for UTI diagnostics, chronic UTI pathophysiology, and treatments. Table 2 displays the frequencies of subcodes mentioned by participants composing each theme, and Figure 2 showcases representative quotes underpinning the major components of each theme.

### 2.4. Theme 1: Physical Health Impacts

When the participants were asked to explain their experiences and greatest challenges dealing with UTIs, many participants described the negative impact of UTIs on their physical health. There were 282 accounts of participants’ experiences with pain and 174 descriptions of genitourinary (GU) symptoms (primarily dysuria, bladder pain, frequency, urgency, and/or incontinence). Aside from pain and GU symptoms, participants also reported that UTIs caused them to experience other symptoms, such as fatigue and malaise, as well as weight gain. In terms of pain, several participants provided in-depth insight into the severity and chronicity of their pain. For example, a respondent from Sweden reflected upon both:
*“[I’ve been] in pain every day [for] 5 years. Some days [are] so bad that I can only cry, and some days it’s more manageable”.**(Sweden, 123597744)*
Some participants cited suffering from severe complications of UTI, such as sepsis (*n* = 3) or hospitalization (*n* = 11). A participant from Canada reported the following: 


*“I was diagnosed [with] having a chronic UTI after I failed to recover from an acute UTI infection in June of 2020, which quickly became relapsing kidney infections, hospitalizations, and turning septic”.*

*(Canada, 123557167)*


### 2.5. Theme 2: Mental Health

#### 2.5.1. Recurrent UTIs Cause or Exacerbate Anxiety and Fear for Many

Participants reported UTI symptoms caused or increased feelings of anxiousness related to locating restroom facilities, participating in activities that make them at higher risk for having a UTI, or having outright fear that they will never experience a reprieve or develop a drug-resistant infection. For example, one participant from Finland wrote,
*“I can hardly ever go away from home, not to speak for a long time. When I go somewhere, I need to find a toilet immediately, so I feel stressed about leaving home”.**(Finland, 123563071)*
Others were afraid or had anxiety about engaging in activities they used to enjoy should they lead to infection. One participant wrote,
*“I love sex, but every time I have it, I am full of anxiety (instead of pleasure and joy) because it feels like a gamble. Am I going to get a UTI this time?”**(Australia, 124115084)*
Participants lived in fear of future infections and the possibility they would never see substantial improvement. One respondent shared,
*“When in periods of little or no pain, I am constantly aware of my bladder and live in terror of another flare”.**(Spain, 123563410)*
Survey participants also reported fear of eventual lack of viable treatment options should they acquire antibiotic-resistant bacteria or a chronic UTI that no longer responds to available treatments: 


*“It makes me afraid of antibiotic-resistant bugs”.*

*(UK, 124016833)*


#### 2.5.2. UTIs Are Very Embarrassing to Some, and Misinformation about the Disease Can Lead to Lowered Self-Esteem

Survey participants described feelings of embarrassment with urinary-related symptoms: 


*“Once a week, I do a hike with a pal. Despite wearing the maximum absorbency pad, I usually have to have a change of clothing when I get home. Once, I leaked on her car seat, which was really humiliating. It makes you feel like an infant”.*

*(UK, 126777479)*


Some commented that UTIs have negatively impacted their self-esteem, and others implied that patients with UTIs are typecast in a way that displaces the disease origin on the patient and plants the seeds of shame and low self-esteem that patients may experience later in life. 


*“Spreading of misinformation when it comes to UTI “facts”, “hints” and “tips”. This is VERY damaging, especially for young girls experiencing their first UTI, and too often sets them up for a lifetime of embarrassment, shame, and UTI misery”.*

*(Australia, 124113868)*


#### 2.5.3. A True Disease and Its Ties to Depression

Many survey participants reported having a chronic and/or recurrent UTI, often accompanied by severe symptoms that also impacted their mental health. Patients described every day as a struggle physically, but also mentally, as they carried the heavy, exhausting burden of living with chronic UTIs.
*“Daily tasks are challenging… swimming upstream every day”.**(US, 123542211)*
Patients also felt a sense of hopelessness—one respondent from the UK shared,
*“… there is the sense that nothing will ever change, and there is no help”.**(UK, 124235774)*
In extreme cases, participants reported feeling so depressed due to sequelae caused by UTIs that they had suicidal ideations. A respondent from the UK wrote,


*“At the age of 23, I feel like my life has ended before its even properly started. Suicide is on my mind most days. Most of the things I enjoyed in life I cannot do. I feel like a shell of the person I used to be”.*

*(UK, 123552632)*


### 2.6. Theme 3: UTIs Severely Limit Many Aspects of Life

#### 2.6.1. UTIs Limit Diet, Exercise, and Productivity and Create a Reliance on Restroom Facilities

Participants described limitations on several aspects of life to prevent, cope, or plan for the unpredictability of UTIs and their corresponding symptoms. To prevent UTIs from occurring, some participants have altered their daily habits and behaviors:
*“I’ve changed my lifestyle and eating habits out of fear of another UTI. No alcohol, less sex, can’t go anywhere without a full water bottle, get anxious when not near a bathroom, etc”.**(US, 124135087)*
Other participants mentioned that UTIs limited their ability to exercise and participate in normal, day-to-day activities: 


*“Living with constant UTIs limits my normal activities due to my need to pee so often. Hiking, shopping, fishing, camping, traveling, and visiting friends are challenging when the proximity of a bathroom is vital”.*

*(US, 124124959)*


UTIs have also impacted some participants’ ability to work or attend school, affecting their financial stability and freedom. A participant from Australia emphasized how UTIs hampered their career: 


*“It has destroyed my career, my ability to work, my fitness, health and mental health, and my ability to earn a high income”.*

*(Australia, 124114670)*


#### 2.6.2. Social Relationships, Sexual Health, and Family Life Impacts

Nearly half of the participants reported that UTIs put constraints on social relationships and family time, placed strain on their intimate-partner relationships, and described its impact on family planning. One Australian participant said,
*“I can hardly go out and socialize with friends and family due to being in constant pain and needing to use the toilet frequently”.**(Australia, 123574965)*
For some participants, the effects on their intimate relationships have been more severe, causing them to abstain from sexual intercourse for 10 years out of fear of getting a UTI. Another Australian participant reflected on being controlled by her chronic UTIs because she cannot prevent them, have intimacy with her husband, and engage with her family.
*“I feel controlled by my chronic UTIs… I grieve the impact on my sex life. I have even wondered whether it would be kinder to my husband to divorce him so he can have a normal sex life with someone. It’s devastating… It put stress on our relationship, and we’ve had many fights about it over the years… I am less engaged in family life”.**(Australia, 124115084)*
Some specifically reported on their experience with UTIs and trying to conceive and even maintain a healthy pregnancy: 


*“The worst part of it all is that the infection caused me to have a late miscarriage”.*

*(UK, 127183812)*


#### 2.6.3. UTIs Are the Main Cause of Decreased Quality of Life 

Many participants pointed to UTIs as the driving force behind their decreased quality of life. The culmination of these limitations has interfered with several aspects integral to participants’ functional, work, familial, and social lives, which had many participants reflect on how their quality of life was drastically reduced to a mere version of itself. One respondent from the UK stated, 


*“It controls every aspect of your life and stops you from living your life. You end up just existing like a shadow of your former self”.*

*(UK, 123565842)*


### 2.7. Theme 4: Treatment and Management Experience

#### 2.7.1. Antibiotic Duration or Dose Is Not Adequate

One of the more prominent areas participants reflected upon was the lack of appropriate antibiotic duration or dose and the refusal of clinicians to prescribe longer treatment courses. For example, in response to naming their greatest hurdle for managing UTIs, this respondent reflected upon their struggles obtaining sufficient treatment to keep their UTIs at bay: 


*“The constant back and forth to doctors, explaining how debilitating UTIs are on my daily life and trying to get more than a 3–5 day antibiotic [course]!! It’s never long enough!!! The UTI cycle just repeats after antibiotics finish”.*

*(UK, 127185976)*


#### 2.7.2. UTI Sufferers Perceived They Received Inadequate Care

In terms of participants’ treatment experiences with clinicians, several reported inadequate care. This included the patient perception that they were not getting the correct treatment and diagnosis (in very general terms) and the reluctance of clinicians to consider emerging research and stayed or unhelpful advice. In terms of receiving proper treatment, one respondent explained that attaining proper treatment was one of their greatest challenges:
*“Getting the correct antibiotic when you need them. It’s well documented in my records the amount I suffer, yet I have to explain and beg for a prescription”.**(UK, 124289918)*
Another commented on the unwillingness of their doctor to examine emerging research:


*“Lack of understanding by doctors and unwilling to look at possible treatment protocols. Doctors refused to look at the research articles I provided and often were offended when I asked questions and sought help”.*

*(Australia, 124126189)*


Other important subthemes that emerged included withholding antibiotics following a negative urine culture despite persistent UTI symptoms (*n* = 21) and being given a diagnosis of interstitial cystitis (*n* = 21). One participant summarized their experience of having persistent symptoms:
*“As a patient, your symptoms are often ignored, and since a lot of infections aren’t even picked up by the current urine tests, you are told, “It’s all in your head”. It’s an absolute disgrace how patients are treated and then sentenced to a life in pain when really, they are suffering from chronic infections”.**(UK, 123599248)*
Several participants mentioned that instead of having a UTI diagnosis, patients are diagnosed with interstitial cystitis.


*“I was told I had [interstitial cystitis] because cultures were negative, but it turns out I had an embedded UTI. MicroGen showed a high bacteria load. It is crazy that I spent years of my life misdiagnosed and getting the wrong treatment”.*

*(US, 123576621)*


#### 2.7.3. Many Participants Reported Negative Interactions with Clinicians

There was an overwhelming number of comments (*n* = 162) detailing that when participants sought care, clinicians dismissed or did not believe their symptoms or experience and lacked empathy towards them or were condescending, particularly when their cultures or urine dipstick tests were negative. One respondent commented,
*“My infection didn’t show on tests, and doctors didn’t believe me. I found this more traumatising than the childhood and domestic violence I’ve been through”.**(UK, 123553386)*
A respondent from the UK commented that the lack of support and understanding from clinicians was the biggest challenge in managing their illness. 


*“The lack of understanding you are given. We are gaslit and undermined, so it turns the illness onto you, and you start questioning your sanity and whether you’re the problem, not the illness. We need more support!”*

*(UK, 123554397)*


#### 2.7.4. Barriers to Accessing Chronic UTI Specialists, Treatments, and Diagnostics

Several participants also reported that accessing a clinician willing to treat them for chronic UTI was a monumental barrier. Two subcategories of accessibility emerged: finding a compassionate doctor able to acknowledge and treat chronic UTIs and getting access to appropriate treatment and diagnostics. Other lesser named but important barriers included physical distances required to see specialists and receiving treatment in a timely manner. Quotes that encapsulate these issues are as follows:


*“Finding a doctor who will prescribe long-term antibiotics. I will have to travel from Ireland to the UK to get treatment”.*

*(Ireland, 123584210)*



*“I spent about 7 years being fobbed off by general practitioners and then it took a year from being referred to a specialist to actually have an appointment”.*

*(UK, 124016545)*


### 2.8. Theme 5: Limited Knowledge and Awareness of UTI Treatment, Burden, and Relationship with Gender

Overall, there was a large number of comments (*n* = 184) describing the medical community as having inadequate knowledge of UTIs and/or awareness of the burden placed on patients, or preconceived beliefs characterizing UTIs as a trivial condition or not taken seriously (*n* = 55). Additional subthemes revolved around clinicians holding thoughts of disbelief (*n* = 13) or lacking knowledge (*n* = 33) of chronic UTIs, as well as the perception that a gender gap accounts for the disparity in UTI treatment and research compared to other diseases (*n* = 52). This participant touched on the perceived lack of knowledge and awareness surrounding UTI treatment when describing his or her greatest UTI management issues: 


*“I have also been given a lot of contradicting information over the years; this makes me wonder if there is a gap in knowledge or just ignorance to this issue”.*

*(UK, 124121525)*


Patients often felt practitioners considered UTIs as trivial illnesses and failed to acknowledge the burden of their condition. This was also reflected by the quantitative poll questions wherein participants rated how others perceived the seriousness of UTIs (scale 0: least important to 100: most important) (median = 20, IQR = 10–30) and the importance of UTIs in their own lives (median = 100, IQR = 90–100) (Figure 3). 

The perception of UTI as a trivial disease was also reflected in the qualitative findings. For example, one participant commented, 


*“Doctors seem to have very little understanding of how crippling they can be” (UK, 123606133) and “underestimate the devastation it brings”.*

*(UK, 124156585)*


A number of participants also shared that clinicians had disbelief or failed to acknowledge the medical diagnosis of chronic UTI. For example, one respondent reflected:


*“They also need to acknowledge chronic UTI—that the bacteria can become embedded in the bladder lining, which stops the antibiotics from working until the bladder lining has been shed”.*

*(Australia, 124118726)*


With UTI considered an infection that primarily impacts females, some participants mentioned or implied that gender gaps account for the disparity in UTI research, treatment, and patient care. One participant from the UK said there is a


*“gender bias in healthcare”, where “women are often not listened to or taken seriously by healthcare professionals, and it is quite frankly killing us”.*

*(UK, 12355917795)*


Another UK participant implied that research, diagnostics, and treatments would be taken more seriously and patients would be listened to if UTIs affected more men than women. When responding to the question on the greatest UTI management challenge, one participant responded,


*“It [a UTI] mostly affects women. Women are not taken seriously, and assumptions are made—pain is in the head, etc. At one point, I was told that “we’d be far more worried if you were a man””.*

*(UK, 12356921739)*


### 2.9. Theme 6: Research Needs

#### 2.9.1. Improvement in Diagnostics for the Detection of UTI Is Imperative

Within the research category, one of the most prevalent needs included updated diagnostics (*n* = 143). Shortcomings in diagnostic testing focused on antiquated tests, lack of sensitivity, specifically the inability to detect fastidious or uncultivable organisms, expediency, and disbelief that laboratories abide by the Kass index to denote meaningful growth. There were also calls for applying polymerase chain reaction technology to detect organisms and using updated technologies when bacteria fail to grow on culture. Some representative quotes regarding cultivation and the Kass index include the following:


*“In [my infectious disease specialist’s] words: urine cultures do not grow everything and can miss infections”.*

*(Canada, 123557167)*



*“Please, we need a change in the way that urine is tested. The current testing is outdated and extremely unpredictable… The laboratory doesn’t culture it long enough. The Kass index is too high, [as the] bacterial load has to be 10,000 [colony forming units (CFU)/mL] to be classed as a UTI. If you come back with 9,999 [CFU/mL], general practitioners say that it’s not a UTI. It’s absolutely ridiculous”.*

*(UK, 123554985)*


Several participants emphasized the need for new diagnostics, especially for patients with chronic UTI. A respondent reflected that the pathophysiology behind this condition does not lend itself to manifest on traditional cultures and commented that care should be symptom based.


*“Review the symptoms of the person, in addition to the lab tests. Please find better lab testing and research to support embedded infection. Doctors need to see that the infection can stay in the body, even if the lab tests are clear”.*

*(US, 123578005)*


#### 2.9.2. Treatment Guidelines Are Hurdles to Care

There were a number of comments describing how these same diagnostics with shortcomings are at the forefront of their country’s UTI treatment guidelines, which hampers their diagnosis (*n* = 52).


*“The NHS [National Health Service] in the UK will only do dip[stick] testing. Most of my symptoms occur, but no infection shows on a dip[stick] test, so I’m told I’m fine. It was only when I learned more and found the right help that they found bacteria after growing it in the lab and further in my bladder wall after a biopsy”.*

*(UK, 123552364)*


Participants also commented that UTI guidelines have a one-size-fits-all approach and put constraints on providers that want to help their patients with persistent UTI symptoms following short courses of therapy. This is reflected in the following quote: 


*“There is an umbrella approach of treatment with short-term antibiotics, which is simply not the answer for many patients who need a long-term solution (for embedded infections, etc.)”.*

*(UK, 123558492)*


Participants also commented that an important shortcoming in their national guidelines was that they lacked specific instructions for chronic UTI treatment and urged researchers to spur change. 

#### 2.9.3. Research of Chronic UTI Pathophysiology and Existence Must Be Undertaken

Survey participants implored researchers to investigate the pathophysiology of chronic UTI (*n* = 52). Participants highlighted many aspects of chronic UTI that must be investigated, with examining and establishing chronic UTI as a diagnosis and its necessity for long-term antibiotics at the forefront (*n* = 17). 


*“There must be a consensus (drawn from current available research and perhaps new studies) that chronic UTI is a real condition (e.g., intracellular bacteria), [and] is distinct from acute UTI (e.g., planktonic bacteria), and cannot be treated, or expected to respond to treatment the same as an acute UTI… No one wants to be on antibiotics for months or years. There must be at once an acceptance that long-term antibiotics are the only thing that appears to help this cohort of patients in any measurable and effective way and a call to action to find better treatment alternatives”.*

*(US, 124505833)*


Other facets of chronic UTI that participants urged researching included its natural disease course, risk factors for development (genetic and others), the relationship between the uro-/microbiome, types of bacterial species implicated, and underlying mechanisms causing chronic UTI, specifically intracellular bacterial pathogenesis and biofilms. Additional clinical research areas included examining the prevalence and reasons for treatment failure among chronic UTI patients and how to distinguish chronic UTI from sporadic UTI.

#### 2.9.4. Validated and Improved Treatments Are Desperately Needed, Especially for Chronic UTI

The most prominent points participants focused on regarding treatment needs revolved around chronic UTI treatment (*n* = 38), finding a cure (*n* = 48), developing better preventive and pain treatments (*n* = 12), and creating more targeted therapies (*n* = 9). Participants also encouraged the study of natural remedies (*n* = 5) and vaccine development (*n* = 4). In terms of chronic UTI treatment, some participants acknowledged that longer treatment durations have been their only option but emphasized that safer and improved therapies are needed due to the side effects of long-term, high-dose antibiotics. Several participants also implored researchers to find a cure to get their lives back: 


*“Please, please find a cure for chronic UTI. It is the most miserable condition which needs to be addressed urgently. I am often contemplating suicide as I feel so awful”.*

*(UK, 124124074)*


#### 2.9.5. Research on the Impact of UTIs on Quality of Life Needs to Be Explored and Disseminated

Participants implored researchers to investigate and understand patients’ lived experiences with UTI, especially its detrimental and sometimes extreme impact on holistic health (*n* = 38). Participants supported disseminating the impact of chronic UTI on quality of life to spread awareness of the seriousness of this less acknowledged disease, gain more footing among clinicians and researchers, and for researchers to understand this unmet medical need. A representative quote is presented below:


*“Ensure you report on the impacts that they have on people’s lives. Medically speaking, they are not serious, but they literally ruin people’s ability to go about daily life as seriously as cancer treatment does”.*

*(UK, 124018170)*


## 3. Discussion

Our Twitter-disseminated survey garnered 466 responses from patients and family members across 22 countries and shed light on the severe physical and mental health impacts of UTIs. We also uncovered shortcomings in patient care stemming from perceived inadequate treatment, negative clinical interactions, and issues in accessing a clinician willing to treat or acknowledge chronic UTI as a diagnosis. Participants also expressed a perceived lack of awareness among clinicians and the public in terms of the gravity of UTI sequelae, reflected in both the qualitative text and quantitative polls, and some rationalized this as a gender gap issue. Lastly, participants highlighted the need for researchers to develop more accurate diagnostics and treatments, particularly geared towards chronic UTI detection and treatment. 

Our study findings in terms of symptomology, lack of antibiotic success, and reports of inadequate care are more in line with those including recurrent or chronic UTI patients [17,18,20]. For example, Flower et al., who analyzed postings from an Internet-based patient support forum hosted by the Cystitis and Overactive Bladder Foundation for women with recurrent UTIs in the UK, reported that some participants attested to constant pain or even experiencing UTI symptoms for decades [17]. The forum also had topics such as “biofilms and chronic infection” and “UTI symptoms but no bacteria found”, which garnered 190 and 4437 views, respectively [17]. There also was an overlap in themes describing atypical and disabling symptoms as: “*symptoms that don’t live in the textbooks*”. and antibiotics that failed to quell their symptoms. Patients interviewed by Hearn et al., who analyzed the impact of chronic and recurrent UTIs on spinal cord injury patients, also described severe symptoms, such as “*felt like my bladder was being torn outside my body*” [20]. Lastly, in Eriksson et al., who interviewed older females primarily with recurrent UTIs, participants reported receiving inadequate care in terms of treatment, follow-up, or physicians nonchalantly addressing their case [18]. 

Other themes, such as the pervasive impacts of UTIs on life, psychological tolls, and lack of support, empathy, and awareness of the UTI burden among clinicians, were detected in studies including patients with either recurrent or sporadic UTIs. For example, a recent systematic review that analyzed 16 qualitative studies found overlap among the following themes: the disruptive nature of UTIs on patients’ lives, their negative impact on quality of life and associated psychological toll, and a lack of acknowledgment and empathy from some clinicians, summarized as “*being heard, seen and cared for with dignity*” [21]. A recent study that interviewed 65 women with sporadic or recurrent UTIs from the US (*n* = 40) or Germany (*n* = 25) also uncovered several limitations that impacted quality of life and strong emotional responses to these limitations and treatment failures, including anxiety, frustration, helplessness, isolation, and embarrassment [16]. Patients also revealed a loss of trust in clinicians or a sense of being unheard [16]. A study that interviewed 29 primarily Caucasian, college-educated females with recurrent UTIs disclosed that participants felt clinicians underestimated and/or dismissed the burden of UTIs on their lives [15]. 

The medical literature often characterizes sporadic UTIs as causing short-term sequelae, remedied by short courses of antibiotics [6]. However, our study introduced a different, more harsh reality, where some participants described disabling pain and long-term symptoms, often termed chronic UTI, coupled with deleterious impacts on their social and psychological well-being. Previous studies that examined experiences for those with recurrent or chronic UTIs did share some overlap in terms of symptom severity and chronicity. Outside of these studies, less emphasis was placed on symptom severity. Similarly, in terms of psychological distress, although Izett-Kay et al. found that UTIs had a negative “impact on quality of life and [an] associated psychological toll”, they did not present the same depth of despair that some of our participants experienced (e.g., depression or suicidal ideation) [21]. 

The stark contrast in symptom severity not matching traditional clinical symptoms, coupled with discussion of lasting symptoms and extreme emotional turmoil, reveals a patient population not well characterized in the medical literature. These patients may represent a subset of individuals that perhaps experience different UTI pathophysiology, along with sequelae that may accompany intracellular bacterial pathogenesis. Murine models of infection have uncovered how uropathogenic *Escherichia coli* (UPEC) exhibit an intracellular life cycle of attachment, reproduction, maintenance within a biofilm-like state, and differentiation into a motile form [22,23,24,25]. This motile form sheds into the bladder lumen, resulting in reinvasion of other bladder cells or bacteriuria [23]. These intracellular reservoirs protect UPEC from neutrophils, while the motile, filamentous forms resist neutrophil engulfment [23]. Murine studies have also found that trimethoprim–sulfamethoxazole failed to eradicate intracellular reservoirs of UPEC even after prolonged (10 days) treatment [26]. Evidence supporting the existence of intracellular bacterial communities (IBCs) and their life cycle in humans has been detected in adults and pediatric patients that have shed IBCs and filamentous UPEC [27,28]. Thus, this intracellular lifecycle may explain the nature of these recalcitrant infections to traditional antibiotic courses. 

As chronic UTI societies shared our survey and 24% of the participants voluntarily characterized their UTIs as recurrent or chronic, this introduced several subthemes and research requests related to this condition. Chronic UTI populations participating in the study may have reflected a large number of instances of the following: inadequate antibiotic therapy (*n* = 92), having negative urine cultures despite persistent symptoms, calls for clinicians to acknowledge this UTI phenotype, and support for more research on chronic UTI pathophysiology and treatment. If human pathophysiology mimics murine models, this would explain why patients may have negative urine cultures despite harboring IBCs, or why short or even extended antibiotic therapies do not mitigate their infection. Taken together, this qualitative analysis that illuminated the severity and chronicity of participants’ UTI experiences, along with emerging research on biofilms and IBCs, warrants clinical researchers to take a deeper look at UTI phenotypes to develop improved treatment and management practices [29].

### Strengths and Limitations

To the best of our knowledge, we were the first to harness the power of social media to distribute our survey using Twitter. Therefore, we had a wide reach and were able to recruit participants from 22 countries, representing patient experiences across different geographical boundaries. However, as most of the participants were from high-income countries in North America and Europe and the remainder were from middle-income countries (1.5%), our results may not fully represent UTI experiences of patients from Asia, Africa, South America, or low- or middle-income countries. As we used an online survey host and the survey was advertised via Twitter, this limited our sample to those with Internet access or to those allowed to access the Internet, as some countries or regions restrict women’s rights. In addition, our survey was only available in English; thus, this limited our responses to participants that could only understand and communicate in English. As Twitter users are on average younger than the general population, we may not have captured UTI experiences more reflective of older adults who experience a higher burden of disease [30,31]. However, responses were somewhat equally dispersed among those under 80 years old, capturing a good distribution of UTI experiences in terms of age.

As we used a survey mechanism, our results are subject to self-reporting bias, wherein claims are not substantiated by a clinical diagnosis, as we did not have access to medical records [32]. However, as our goals were to better understand participants’ experiences across large geographical spaces, it would not be feasible to garner this information as it would require access to individual medical records across many healthcare systems. Another shortcoming of this method is that we were unable to probe to have a further explanation for nuanced terms, such as “embedded” UTI and had to make inferences based on the information provided. The last weakness of this survey method is that those with more severe UTI cases may have had more motivation to participate to instill change; in addition, certain chronic UTI groups “retweeted” our survey, which may have selected more chronic UTI sufferers than are present in the mainstream population. Nonetheless, our survey results draw attention to a patient group that feels neglected, undertreated, and unheard by the medical establishment. By using this anonymous survey mechanism, we were able to foster a forum for these patients to freely share their perspectives. This might not have been the case if typical interviews were conducted, as they may incur social desirability bias or reflexivity—the influence of the researcher on participant responses [32,33]. 

In terms of our methods, the use of four independent coders enhanced theory triangulation, bringing together different perspectives (anthropology, biology, nursing, microbiology, psychology, and sociology) to extract all possible nuances from the text [34]. We also ensured the validity by coding in duplicate and evaluating, tracking, and adjudicating codes with kappa scores < 0.7; when that was the case, discussion and subsequent adjudication were performed by the coding team to ensure investigator triangulation [34]. We also performed methodological triangulation by complementing our qualitative findings that revealed a lack of awareness of UTI seriousness with a quantitative poll [34,35].

## 4. Materials and Methods

### 4.1. Survey Design and Survey Questions 

We conducted an exploratory study to better understand individuals’ experience with UTI and elicit research suggestions via a Twitter-disseminated survey promoted by the Urinary Tract Infection Global Alliance (UTIGA) between January and June 2021. The survey targeted individuals that had experienced a UTI or had a relationship with an individual with a history of UTIs (i.e., family members or spouse) and healthcare providers. The survey was shared (i.e., tweeted) on the UTIGA Twitter account on 25 January 2021, and was posted on the UTIGA website (Figure 4). Any UTIGA Twitter follower or other Twitter user could see the tweet, elect to participate by clicking the survey link, and share the tweet (i.e., retweet). All Twitter analytics were collected at the close of the study (5 June 2021), including the number of impressions (i.e., how many times Twitter users viewed the tweet), likes (i.e., how many times a person “liked” the tweet), and retweets.

The mixed-methods survey consisted of five questions, including three qualitative and two quantitative questions. The qualitative questions aimed to understand a person’s experience with UTIs, including how their life had been affected by UTIs, their greatest hurdle in managing UTIs, and what they would say to a researcher studying them (Figure 5A). The quantitative questions aimed to measure the burden that UTIs have on participants’ lives and how they perceived society views UTIs by asking participants to rate on a scale of 1 to 100 (highest) (1): How seriously do you think UTIs are taken by others? (2) How important are UTIs in your life? 

In the initial survey, we also collected data on country of residence and asked the participants to voluntarily list their email addresses. We sent a follow-up email inquiring about demographic characteristics, including age range (in decades), gender identity, race, ethnicity, and relationship with UTIs (patient, clinician, caregiver, researcher). As our survey was advertised via Twitter, we were not able to collect information on those that did not choose to respond, and participants could take the survey in any setting. All survey responses were handled confidentially, and materials and methods were approved by the Nationwide Children’s Hospital IRB (study number: 00001773).

### 4.2. Qualitative Analyses of Participants’ Responses

The Baylor College of Medicine qualitative research team consisted of four coders (M.V.-K., L.L., K.O., and C.H.-M.) trained in qualitative research methods. Figure 5B provides an overview of the approach used for coding and thematic analysis. The research team used an inductive coding approach [36]. To develop the initial codebook, the team free-coded the first 100 responses across the qualitative questions: Q1, Q2, and Q3. For the remainder of the responses, the team coded in duplicate and met weekly to discuss the codes, refine the codebook if necessary, and adjudicate responses if there were discrepancies in the agreement between the coders (i.e., if the kappa score was <0.7 for an individual code). See Appendix A for the codebook. Data saturation was achieved well before all 466 responses were analyzed. Once all three questions were coded and adjudicated for all 466 participants, the team conducted a thematic analysis, grouping the coded text responses into themes and subthemes that best represented the patterns of responses (Figure 5B). The research team used member checking to agree that the final themes accurately captured the participants’ responses. All data management, storage, and qualitative analyses were conducted using the NVivo software versions 1.6.1. and 1.6.2. When quoting specific participants in this paper, we used a unique identifier to keep the participants anonymous. The identifier included the participant’s country of residence and a unique, nine-digit identifier.

## 5. Conclusions

Our social-media-promoted survey attracted a sizeable number of participants wherein a number of individuals detailed the extreme physical and mental health impacts of UTIs. Subpopulations that described their UTIs as chronic or recurrent often encountered difficulty receiving adequate treatment to quell their symptoms and reported feeling dismissed by healthcare providers, as their infections did not manifest on standard urine cultures. They felt that the burden of their illness was not recognized by clinicians or the public and suggested that researchers focus on further understanding chronic UTIs, developing better diagnostics, and validating and improving treatment options that have fewer side effects than long-term antibiotics. This report on patient experiences challenges the norms that UTIs share textbook sequelae and can be remedied by short courses of antibiotics. Future research investigating this more severe UTI phenotype needs to be undertaken to establish a consensus definition to study its epidemiology, thus providing a foundation to evaluate new diagnostics and treatments for this group of patients with unrequited pleas for care.

## Figures and Tables

**Figure 1 antibiotics-11-01687-f001:**
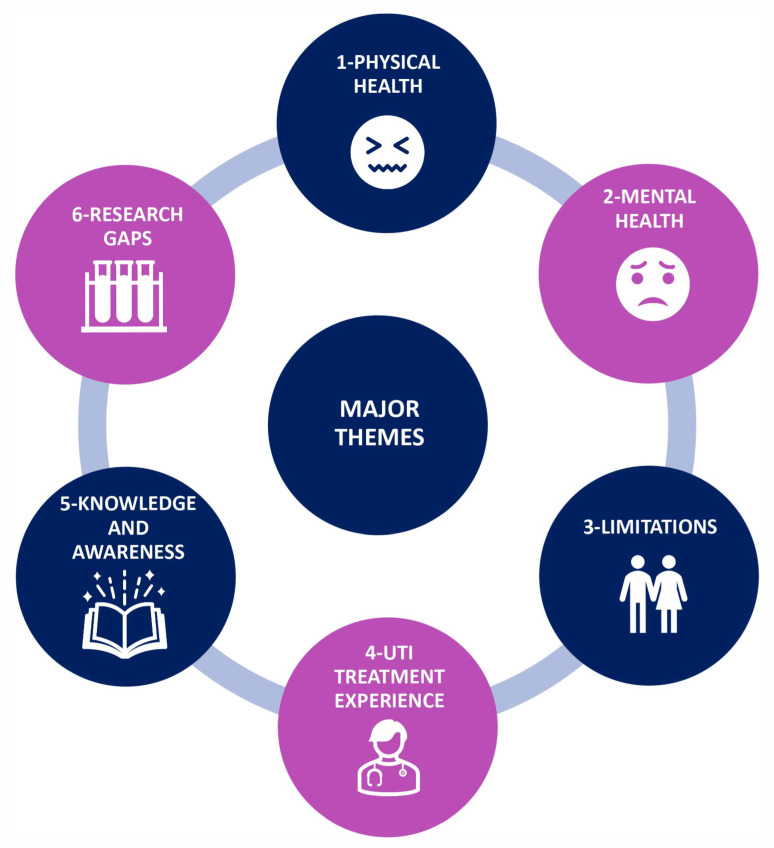
Major themes derived from thematic analysis.

**Figure 2 antibiotics-11-01687-f002:**
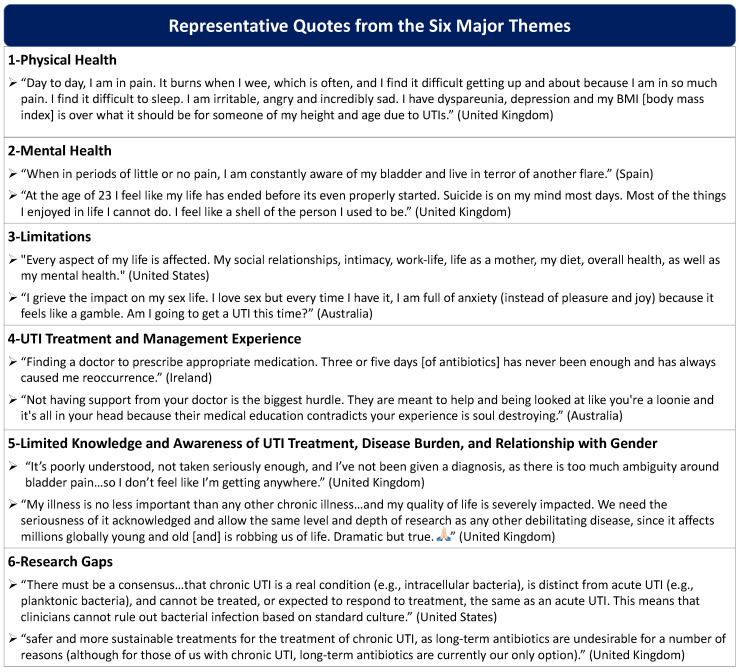
Representative quotes from six major themes.

**Figure 3 antibiotics-11-01687-f003:**
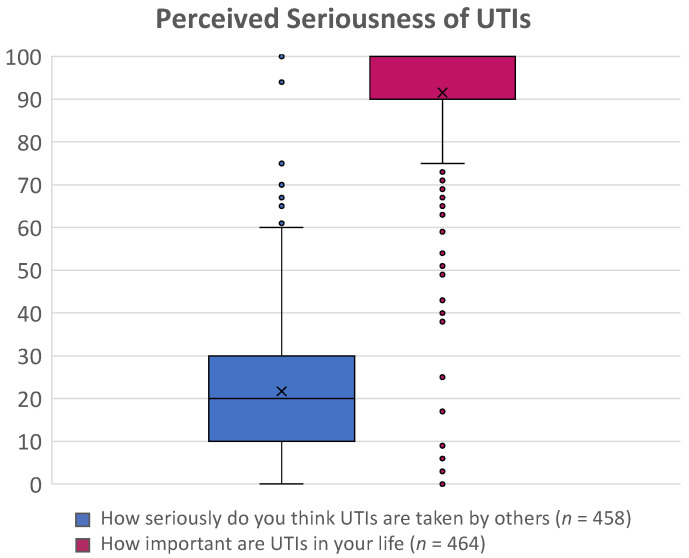
Descriptive statistics from the survey poll. Box plot whiskers represent the 0.025 and 0.975 percentiles, the “X” denotes the mean, and the solid dots represent outliers.

**Figure 4 antibiotics-11-01687-f004:**
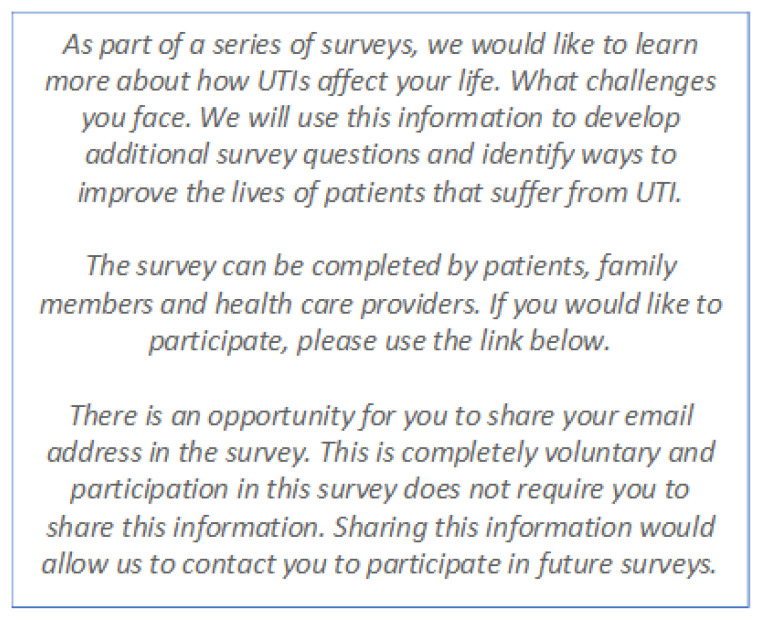
Participant recruitment statement posted to the UTIGA website.

**Figure 5 antibiotics-11-01687-f005:**
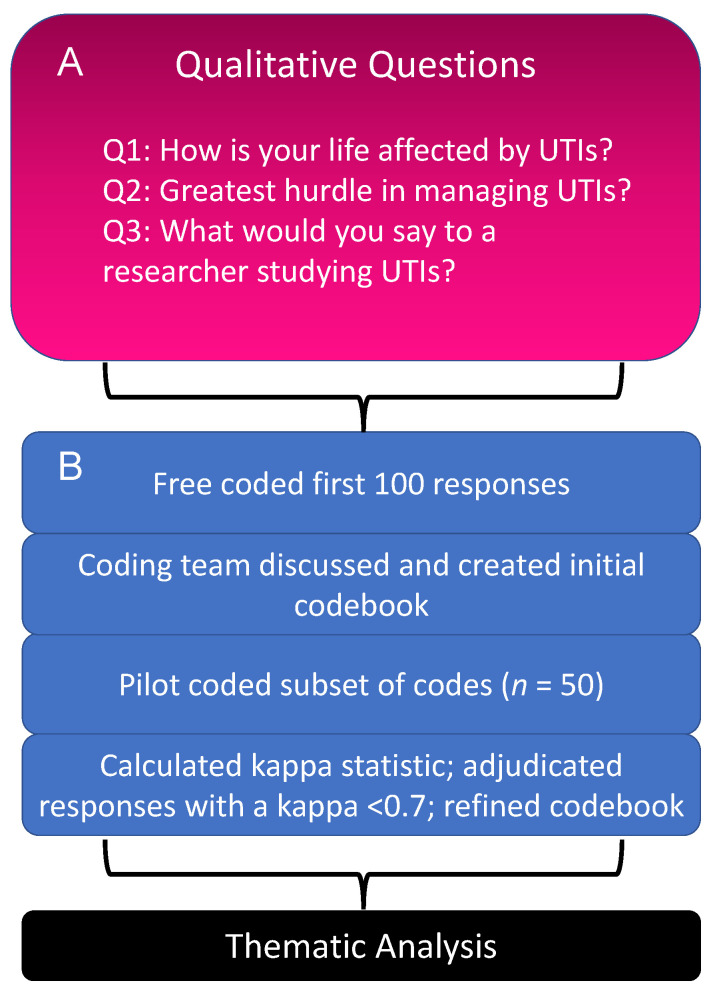
(**A**) Qualitative questions proposed to survey participants. (**B**) Overview of the qualitative analysis process undertaken for each qualitative question (Q1–Q3).

**Table 1 antibiotics-11-01687-t001:** Characteristics of the survey participants ^a^.

Gender	Number (%)
Female	200 (96.6)
Male	7 (3.4)
**Age Range**	
18 to 40	57 (27.5)
41 to 60	80 (38.6)
61 to 80	69 (33.3)
>80	1 (0.5)
**Race**	
White	196 (94.7)
Other ^b^	11 (5.3)
**Participant Region**	
United Kingdom/Ireland	257 (55.2)
United States/Canada	127 (27.3)
Oceania ^c^	52 (11.2)
Europe	25 (5.4)
Other ^d^	5 (1.2)

^a^ A total of 207 out of 466 participants provided demographic information. ^b^ Other includes Asian (*n* = 2), Black (*n* = 1), Multiracial (*n* = 4), Middle Eastern (*n* = 1), Hispanic (*n* = 1). ^c^ Australia (*n* = 45), New Zealand (*n* = 7). ^d^ Pakistan, United Arab Emirates, India, Vietnam. Bolded text represents the title of each descriptive category.

**Table 2 antibiotics-11-01687-t002:** Frequency of subcodes mentioned by participants comprising the six major themes.

Theme and Subcodes	No.	Theme and Subcodes	No.
**Physical health**		**UTI treatment and management**	
General pain	282	Antibiotic dose or duration not adequate	92
Chronic pain	89	Perception of inadequate care	129
Severe pain	54	Negative clinician interactions	162
Genitourinary symptoms	174	Accessing care:	
Fatigue	27	Chronic UTI specialist or treatment	85
Malaise	15	Travel	12
**Mental health**		Timeliness	15
General mental health	66		
Anxiety	72	**Limited knowledge and awareness of UTI treatment, burden, and relationship with gender**	
Depression	50		
Distress/stress	32	Among medical communities	184
Embarrassment	11	UTIs considered trivial	55
Fear	41	Disbelief or knowledge of chronic UTI	46
Frustration	45	Gender gap	52
Suicidal ideation	29		
**Limitations**		**Research gaps**	
Quality of life	128	Improved diagnostics	143
Overall functioning	113	Treatment guidelines as hurdles to care	52
Diet	38	Prevention	20
Exercise	40	Cure	48
Sleep	40	Quality of Life	38
Work, school, or economic	148	Chronic UTI:	
Unpredictability of UTIs	56	Pathophysiology	52
Dependency ^a^	107	Treatment	38
Social health and family life	191	Establish as a medical diagnosis	17
Sexual health and intimacy	69		
Family planning ^b^	16		

^a^ Requiring proximity to the restroom or medical intervention. ^b^ Challenges, losses, or inability to have children. No.—number. Bolded text represents the major themes derived from our analysis.

## Data Availability

Data from the codebook is provided in Appendix A, and representative quotes are outlined in the manuscript. De-identified qualitative transcript data are available upon request from the corresponding author due to privacy.

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
