# Peer review of "Qualitative Analysis of a Twitter-Disseminated Survey Reveals New Patient Perspectives on the Impact of Urinary Tract Infection"

_antibiotics, 2022, doi:10.3390/antibiotics11121687_

Round 1

Reviewer 1 Report

I found the study very interesting. I have some little recommendations:

1.      The authors could comment if the questionnaire is previously validated or this is a pilot study.

2.      Мy personal opinion is that more results should be tabulated or the most common answers should be summarized in a table (depending on the type of questions, as the authors do not indicate how many open questions there are). A tabular presentation will present the results much more clearly and it will be easier for the reader to draw the relevant conclusions correctly.

3.      Limitation of the study is the number of participants in age group using Twitter and involved in the study. There is a possibility to more spread of UTI infections in older people (over 60 and 80 years), but they do not use this communication channel (from my point of view in my country, the authors could comment their opinion).

Author Response

Reviewer 1

I found the study very interesting. I have some little recommendations:

  1. The authors could comment if the questionnaire is previously validated, or this is a pilot study.

Thank you for your question. We did not consider this work a pilot study but rather exploratory in nature. As such, we created survey questions that were broad in scope to generate hypotheses. Since our study was of this exploratory nature, we were not aiming to measure a specific construct, so we did not previously use or validate these questions.

We clarified the intention of our study by adding the following underlined text when introducing our study methods in lines 571-573:

We conducted an exploratory study to better understand individuals’ experience with UTI and elicit research suggestions via a Twitter-disseminated survey promoted by the Urinary Tract Infection Global Alliance (UTIGA) between January and June 2021.

  1. Мy personal opinion is that more results should be tabulated or the most common answers should be summarized in a table (depending on the type of questions, as the authors do not indicate how many open questions there are). A tabular presentation will present the results much more clearly and it will be easier for the reader to draw the relevant conclusions correctly.

Thank you for this comment. In the manuscript, we explained we used three open-ended questions in the Methods in lines 583-586, and that we derived themes 1-5 primarily from open-ended questions 1 and 2 and theme 6 from open-ended question 3 in lines 141-145 in the Results. We created a new Table 2, which tabulates the frequency of the most common subcodes comprising each of the major 6 themes. We added an additional Figure (Figure 2) that summarizes that contains representative quote(s) for each theme to help the reader better conceptualize our findings.

  1. A limitation of the study is the number of participants in age group using Twitter and involved in the study. There is a possibility to more spread of UTI infections in older people (over 60 and 80 years), but they do not use this communication channel (from my point of view in my country, the authors could comment their opinion).

We agree that this is a limitation; thus, we may not have captured UTI experiences more reflective of older adults who experience a higher UTI burden. However, we did collect responses almost equally across all age ranges, including those between 60 and 80 years old. We added a sentence detailing this in the limitations in lines 539-543 as follows:

“As Twitter users are on average younger than the general population, we may not have captured UTI experiences more reflective of older adults who experience a higher burden of disease [1,2]. However, responses were somewhat equally dispersed among those under 80 years-old, capturing a good distribution of UTI experiences in terms of age.”

References

  1. Wojcik, S.; Hughes, A. Sizing up Twitter Users; Pew Research Center: 2019.
  2. Schmiemann, G.; Kniehl E Fau - Gebhardt, K.; Gebhardt K Fau - Matejczyk, M.M.; Matejczyk Mm Fau - Hummers-Pradier, E.; Hummers-Pradier, E. The diagnosis of urinary tract infection: a systematic review. 2010.

Reviewer 2 Report

The authors ask some questions about the impact and experience of urinary tract infection in social media, obtaining most of the answers from patients with recurrent or chronic urinary tract infection, with complaints about the misunderstanding of their usual doctors and the need for more studies for their diagnosis and treatment.

Urinary tract infection, as the authors point out, is a very common infection, which especially affects women and is usually easily resolved. However, there are some populations in which it can lead to a serious infection, either due to morbidity and mortality or because of the impact on quality of life, especially when the episodes are frequent or chronic.

The introduction is short, understandable and adequately places the objective of the study in its scientific and social context.

The results are long, repetitive and tedious, they seem like the enumeration of a series of anecdotes collected online. More graphic information should lighten the results.

The study design should be explained in more detail. A serious descriptive observational study is more than posting a question on twitter and collecting anecdotes.

As the authors point out in the limitations, the selection bias has a significant burden in this work, since people who have had bad experiences or a significant impact on their quality of life are much more motivated to answer. The homogeneity of the sample also stands out, being difficult to extrapolate the results to other sociocultural contexts outside of a medium-high social stratum of Anglo-Saxon population.
As the authors also point out, the diagnosis of the people who answer the survey is uncertain, as there are many pathologies with LUTS.
It is striking too that of the 466 initial responses, less than half responded later.

Some of the people who respond point out that their usual doctors "are determined" to deny the diagnosis of chronic urinary tract infection. I believe that all of us clinicians have had patients who get stuck on a misdiagnosis because they have been "informed" on the internet, but lack the training and education to interpret much of the information (or misinformation) online.

Author Response

Reviewer 2

The authors ask some questions about the impact and experience of urinary tract infection in social media, obtaining most of the answers from patients with recurrent or chronic urinary tract infection, with complaints about the misunderstanding of their usual doctors and the need for more studies for their diagnosis and treatment.

Urinary tract infection, as the authors point out, is a very common infection, which especially affects women and is usually easily resolved. However, there are some populations in which it can lead to a serious infection, either due to morbidity and mortality or because of the impact on quality of life, especially when the episodes are frequent or chronic.

The introduction is short, understandable and adequately places the objective of the study in its scientific and social context.

1. The results are long, repetitive and tedious, they seem like the enumeration of a series of anecdotes collected online. More graphic information should lighten the results.

We modified Figure 1 to only introduce the six major themes and created a new Figure 2 that displays representative quotes from the participants. We additionally removed two introductory paragraphs that replicated the more in-depth exploration of our findings and dropped 17 quotes in tracked changes.

2. The study design should be explained in more detail. A serious descriptive observational study is more than posting a question on twitter and collecting anecdotes.

Our study was exploratory in nature, as no prior studies had queried the online UTI community for their experiences or needs. As such, we created survey questions that were broad in scope to generate hypotheses and used qualitative research methods to analyze these results. We clarified the intention of our study by adding the following underlined text when introducing our study methods in lines 571-573:

We conducted an exploratory study to better understand individuals’ experience with UTI and elicit research suggestions via a Twitter-disseminated survey promoted by the Urinary Tract Infection Global Alliance (UTIGA) between January and June 2021.

We provided relevant details of the qualitative analysis according to Tong et al., which sets a standard for evaluating qualitative research. These included details on study design and detailed methods on our coding analysis, interrater reliability, as well as the adjudication process that are in line with qualitative research methods [1]. We added additional details in the Methods regarding the study design per Tong et al. as follows:

Lines 598-600: “As our survey was advertised via Twitter, we were not able to collect information on those that did not chose to respond, and participants could take the survey in any setting.”

Line 612: “Data saturation was achieved well before all 466 responses were analyzed.”

  1. As the authors point out in the limitations, the selection bias has a significant burden in this work, since people who have had bad experiences or a significant impact on their quality of life are much more motivated to answer.

Yes, we do mention selection bias in our Limitations in lines 544-546. However, it does reveal a segment of the population that suffers a significant burden from lower urinary tract symptoms and do not feel heard by the medical establishment. We derived several themes from this large sample of respondents, especially by qualitative standards. Our novel findings contribute to the qualitative literature, which should spur dialogue and awareness regarding this population with unmet medical needs from current treatment standards. We mention in the conclusion that further research should be undertaken to characterize and understand the prevalence of this subtype of lower urinary tract symptoms.

  1. The homogeneity of the sample also stands out, being difficult to extrapolate the results to other sociocultural contexts outside of a medium-high social stratum of Anglo-Saxon population.

Correct, as noted in the Results and Limitations, the majority of our respondents were White and from higher-income countries across North America, the UK, Oceania, and Europe. Thus, our study reflects the experiences of individuals within these countries, as mentioned in the Limitations. However, the goal of qualitative research is to perform an in-depth exploration of a topic, not to generalize our findings to a larger group, in contrast to quantitative research [2]. 

  1. As the authors also point out, the diagnosis of the people who answer the survey is uncertain, as there are many pathologies with LUTS.

Yes, we mention this in the Limitations in lines 544-546. This does reveal a subset of those with more severe lower urinary tract symptoms that sets the stage for further inquiry to better understand the pathology behind these patients.

  1. It is striking too that of the 466 initial responses, less than half responded later.

We do not think this is surprising, as we had a time gap of over one year between the initial time the survey was available (Jan 2021-June 2021) to when we emailed participants a link to our follow-up demographics survey (September 2022). As we also contacted the respondents via email (instead of a broad call via Twitter), the demographic survey request/link may not have reached their inbox (e.g., possibly entered a spam folder), limiting replies. Additionally, in the initial survey, not all respondents provided their email address for follow-up, limiting the number of potential responses to 448. We added this detail in line 100 in the Results section.

7. Some of the people who respond point out that their usual doctors “are determined” to deny the diagnosis of chronic urinary tract infection. I believe that all of us clinicians have had patients who get stuck on a misdiagnosis because they have been “informed” on the internet, but lack the training and education to interpret much of the information (or misinformation) online.

We acknowledge that there may be misinformation available online and that patients can and do seek some information from non-evidenced-based websites. However, the large number of responses we received from this study revealed that these patients feel physical pain and immense discomfort related to their UTIs. As researchers and clinicians, there is a real need to share these experiences and have open conversations with patients about these concerns. We believe clinicians and patients may benefit the most by having this open dialogue moving forward.

References

  1. Tong, A.; Sainsbury, P.; Craig, J. Consolidated criteria for reporting qualitative research (COREQ): a 32-item checklist for interviews and focus groups. Int J Qual Health Care 2007, 19, 349-357, doi:10.1093/intqhc/mzm042.
  2. Carminati, L. Generalizability in Qualitative Research: A Tale of Two Traditions. Qualitative Health Research 2018, 28, 2094-2101, doi:10.1177/1049732318788379.

Reviewer 3 Report

This survey report is descriptive and well-constructed emphasizing the problem faced by UTI patients. These types of survey data need to be collected more often to involve patients to include their perspectives while designing new therapies and treatments.

I have the following minor comments about the study:

  • This article is more appropriate for a survey report category rather than a research article. Because it includes the patient's view and their experiences related to UTIs. While keeping it under the research article category would confuse the readers. 
  • In the Abstract section, the term UTIs is used more frequently. It would be great to replace it with an 'article'. The frequent use of any word in an article impacts the reader's interest. 
  • While talking about the severe impact of UTIs, there is still available treatment that requires discussion while providing constructive suggestions for improvement. Here the problem is addressed in sufficient detail however in-depth discussion to deal with the problem would be more appreciated.
  • This survey data look biased toward the problem and reflect that the scientific community is not giving its best to address the issue. There are drugs and certain antibiotics approved by the FDA (Nitrofurantoin is one of them), with few in the pipeline. It would be great to include that section to spread awareness among patients.
  • Along with this adopting a precautionary mindset (hygiene and cleanness) could significantly decrease the chances of infection and avoids disease severity. The awareness among the patients could be pivotal in breaking the tabu associated with UTIs.
  • One limitation of this survey was that it did not include patients from the African continent and South East Asia regions where the literacy rate is low and the Middle Eastern regions where women are still fighting for their rights.
  • The author described that clinicians do not prescribe long antibiotic treatments. The prime reason for this is that drug abuse and excessive and continuous use of antibiotics leads to antibiotic resistance that could compromise the patient health and increase the chances of co-infection with other organisms. In this case, an efficient short treatment could be the best approach and ongoing research will help in overcoming that type of issue.

Round 2

Reviewer 2 Report

1. The results are long, repetitive and tedious, they seem like the enumeration of a series of anecdotes collected online. More graphic information should lighten the results.

We modified Figure 1 to only introduce the six major themes and created a new Figure 2 that displays representative quotes from the participants. We additionally removed two introductory paragraphs that replicated the more in-depth exploration of our findings and dropped 17 quotes in tracked changes.

With the modifications and introduction of the Figures, the results are lighter and more amenous for the reader.

2. The study design should be explained in more detail. A serious descriptive observational study is more than posting a question on twitter and collecting anecdotes.

Our study was exploratory in nature, as no prior studies had queried the online UTI community for their experiences or needs. As such, we created survey questions that were broad in scope to generate hypotheses and used qualitative research methods to analyze these results. We clarified the intention of our study by adding the following underlined text when introducing our study methods in lines 571-573:

 We conducted an exploratory study to better understand individuals’ experience with UTI and elicit research suggestions via a Twitter-disseminated survey promoted by the Urinary Tract Infection Global Alliance (UTIGA) between January and June 2021.

We provided relevant details of the qualitative analysis according to Tong et al., which sets a standard for evaluating qualitative research. These included details on study design and detailed methods on our coding analysis, interrater reliability, as well as the adjudication process that are in line with qualitative research methods [1]. We added additional details in the Methods regarding the study design per Tong et al. as follows:

Lines 598-600: “As our survey was advertised via Twitter, we were not able to collect information on those that did not chose to respond, and participants could take the survey in any setting.”

Line 612: “Data saturation was achieved well before all 466 responses were analyzed.”

The study design after the corrections is explained with the enough detail.

   3. As the authors point out in the limitations, the selection bias has a significant burden in this work, since people who have had bad experiences or a significant impact on their quality of life are much more motivated to answer.

Yes, we do mention selection bias in our Limitations in lines 544-546. However, it does reveal a segment of the population that suffers a significant burden from lower urinary tract symptoms and do not feel heard by the medical establishment. We derived several themes from this large sample of respondents, especially by qualitative standards. Our novel findings contribute to the qualitative literature, which should spur dialogue and awareness regarding this population with unmet medical needs from current treatment standards. We mention in the conclusion that further research should be undertaken to characterize and understand the prevalence of this subtype of lower urinary tract symptoms.

   4. The homogeneity of the sample also stands out, being difficult to extrapolate the results to other sociocultural contexts outside of a medium-high social stratum of Anglo-Saxon population.

Correct, as noted in the Results and Limitations, the majority of our respondents were White and from higher-income countries across North America, the UK, Oceania, and Europe. Thus, our study reflects the experiences of individuals within these countries, as mentioned in the Limitations. However, the goal of qualitative research is to perform an in-depth exploration of a topic, not to generalize our findings to a larger group, in contrast to quantitative research [2].

  5.  As the authors also point out, the diagnosis of the people who answer the survey is uncertain, as there are many pathologies with LUTS.

Yes, we mention this in the Limitations in lines 544-546. This does reveal a subset of those with more severe lower urinary tract symptoms that sets the stage for further inquiry to better understand the pathology behind these patients.

The limitations of the study are explained in the manuscript.

    6 It is striking too that of the 466 initial responses, less than half responded later.

We do not think this is surprising, as we had a time gap of over one year between the initial time the survey was available (Jan 2021-June 2021) to when we emailed participants a link to our follow-up demographics survey (September 2022). As we also contacted the respondents via email (instead of a broad call via Twitter), the demographic survey request/link may not have reached their inbox (e.g., possibly entered a spam folder), limiting replies. Additionally, in the initial survey, not all respondents provided their email address for follow-up, limiting the number of potential responses to 448. We added this detail in line 100 in the Results section.

With the addition of this detail in line 100 in the Results, it is more understandable.

7. Some of the people who respond point out that their usual doctors “are determined” to deny the diagnosis of chronic urinary tract infection. I believe that all of us clinicians have had patients who get stuck on a misdiagnosis because they have been “informed” on the internet, but lack the training and education to interpret much of the information (or misinformation) online.

 We acknowledge that there may be misinformation available online and that patients can and do seek some information from non-evidenced-based websites. However, the large number of responses we received from this study revealed that these patients feel physical pain and immense discomfort related to their UTIs. As researchers and clinicians, there is a real need to share these experiences and have open conversations with patients about these concerns. We believe clinicians and patients may benefit the most by having this open dialogue moving forward.

As the authors suggest, open conversations with patients about these concerns may help the process of healing.Tissue and cell damage may remain after the resolution of the infectious process through epigenetic mechanisms. The sensations described by the patients may be an expression of this damage. However, the effect of antibiotics after the resolution of the infectious process is doubtful, beyond the placebo effect. The appearance of multiresistant strains after the administration of short courses of antibiotic therapy is a danger that the clinician cannot ignore.
